# Immersive Multimodal Translation: A Proxy Task for Cross-modal and Objective Evaluation of Unified Models

## Abstract

Unified multimodal models that jointly perform image understanding and generation have achieved substantial progress. However, a critical challenge persists in establishing rigorous evaluation protocols. Existing benchmarks typically assess generation and understanding tasks independently and rely on large multi-modal language models (MLLMs) for scoring. Such approaches introduce language-centric biases and lack objective ground truth, thereby limiting the reliability and fairness of model assessment. To address this, we propose Immersive Multi-modal Translation (IMT), a novel proxy task that requires models to translate textual content within images while preserving visual context. IMT naturally captures cross-modal synergy between understanding and generation, while enabling transparent, objective evaluation through established metrics from natural language processing and computer vision. To support systematic study, we construct IMTBench, a benchmark spanning three scenarios, including document, webpage, and scene image, with nine languages, and 2,000 carefully curated samples. IMTBench incorporates a three-dimensional evaluation framework measuring translation quality, background fidelity, and visual text rendering accuracy. Extensive experiments across diverse unified multi-modal architectures reveal that current open-source models still fall significantly short of commercial expert systems. By providing objective, cross-modal evaluation protocols, we believe that IMT and IMTBench can offer actionable guidance for future research in unified multi-modal intelligence.

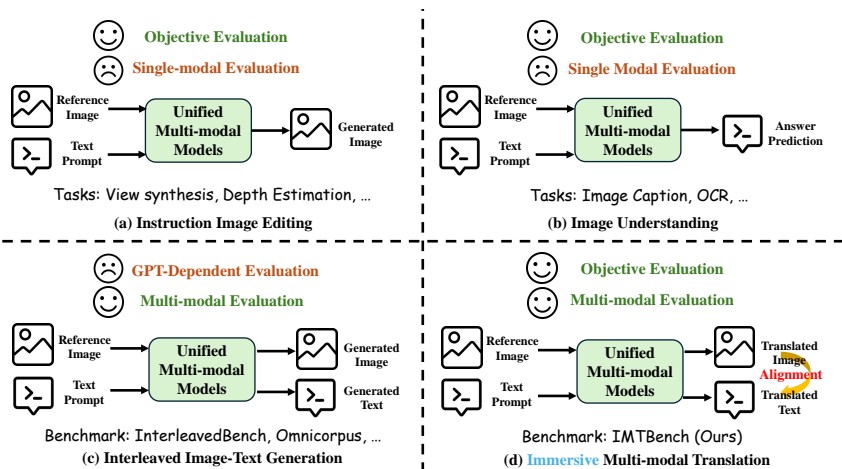

Figure 1: Comparison of existing evaluation tasks for unified multimodal generation and understanding models and our proposed Immersive Multi-modal Translation (IMT) task. Unlike prior tasks, IMT simultaneously supports objective evaluation and cross-modal assessment. The term "Immersive" emphasizes that the generated text and images must remain aligned in between text and image.

# 1 INTRODUCTION

Recent advances in image understanding and generation have fueled growing interest in unified multi-modal models that jointly handle both modalities. A range of frameworks have been proposed, spanning proprietary systems such as GPT-Image-1 (OpenAI, 2025) and Banana-nano [1], and open-source efforts including Qwen-Image (Wu et al., 2025a), Bagel (Deng et al., 2025), Blip-3o (Chen et al., 2025a), and UniWorld (Lin et al., 2025a). These systems employ diverse paradigms, including autoregressive, diffusion-based, cascaded, hybrid, and have achieved notable success across tasks such as image generation, editing, and understanding. However, despite rapid progress, it remains unclear which paradigm offers the most promise for general-purpose multimodal intelligence. Addressing this question critically depends on how we evaluate such models.

As shown in Fig. 1(a) and (b), current practice typically assesses generation and understanding tasks (Downs et al., 2022; Silberman et al., 2012; Liu et al., 2024c;b) in isolation, neglecting the original motivation for unification: to achieve synergy between cross-modal comprehension and contextual generation. In addition, most benchmarks for image-text generation (Xiao et al., 2025; Liu et al., 2024a) and editing (Ye et al., 2025; Liu et al., 2025) rely heavily on large multimodal language models (MLLMs) such as GPT-4o (OpenAI, 2024) for scoring Fig. 1(c). While such models provide valuable reference signals, they introduce two major limitations: (1) Many generation or understanding task can only evaluate performances only in a single modality, and (2) their judgments depend on Multi-modal Large Language Model (MLLM), which is influenced by pretraining data. So they cannot be regarded as truly objective metrics, undermining the credibility of evaluation. As a result, existing frameworks fail to provide a fair and rigorous assessment of unified models.

To address these limitations, we introduce Immersive Multimodal Translation (IMT), a novel proxy task designed to evaluate unified image generation and understanding models. Illustrated by Fig. 1(d), IMT requires models to produce text–image aligned translations: given an image containing text and a target language, the model generates a faithful translation seamlessly integrated into the visual context. Crucially, IMT captures both objective evaluation and cross-modal synergy and decomposes into three subtasks: (1) translating the textual content, (2) rendering the translated text in the image, and (3) preserving the original background and layout. Each subtask aligns with established problems in NLP or computer vision, enabling validated metrics for transparent, large-model-independent assessment. Beyond methodological rigor, IMT supports practical applications in tourism, education, workplace collaboration, and social communication.

To facilitate systematic study, we construct IMTBench, a benchmark for evaluating unified multi-modal models on IMT. IMTBench spans three scenarios, including documents, webpages, and scene images, with nine languages, comprising 2k carefully annotated samples. Building on insights from translation and vision research (Rei et al., 2020; Zhang et al., 2018), we design a three-dimensional evaluation protocols covering three subtasks above mentioned, covering visual, text and alignment score. Extensive experiments on IMTBench with commercial pipelines and unified multi-modal models reveal that existing models still lag behind expert systems in multi-modal translation, highlighting substantial room for improvement and structural trade-offs among paradigms. Furthermore, we fine-tune on IMT-1M, which are curated with the same pipeline as IMTBench, substantially boosts model performance on IMT task. Several fine-tuning observations are reported in this work, with the aim of providing insights for the community on training unified generation and understanding models. Our main contributions are summarized as follows:

- We propose **Immersive Multimodal Translation (IMT)** as a new proxy task for evaluating unified image generation and understanding models, alleviating the subjectivity and bias issues of prior evaluation methods that rely heavily on large-model–based scoring.

- We introduce **IMTBench**, a benchmark spanning a three-dimensional evaluation protocol covering (i) cross-modal contextual comprehension, (ii) background coherence in context-aware image editing, and (iii) semantic–visual synergistic generation.

- Through extensive evaluations of both open-source and commercial unified multimodal architectures, we reveal substantial performance gaps in immersive muti-modal translation tasks quantitatively. External fine-tuning experiments on IMT-1M uncover the critical structural trade-offs across different paradigms.

---

[1]aistudio.google.com/models/gemini-2-5-flash-image

## 2 RELATED WORKS

### 2.1 UNIFIED MULTI-MODAL UNDERSTANDING AND GENERATION MODELS

Recent research has increasingly focused on unified multi-modal architectures that integrate image understanding and generation within a single framework. According to the decoding paradigm (Zhang et al., 2025a), we categorize these unified multi-modal models into three categories: diffusion-based, auto-regressive-based, and hybrid-based methods. Diffusion-based extend diffusion models to multi-modal generation. Dual Diffusion introduces dual-branch denoising for text and image latents with cross-modal attention. UniDisc (Swerdlow et al., 2025) unifies modalities in a discrete token space, while FUDOKI (Wang et al., 2025a) replaces timestep-based diffusion with discrete flow matching for better global reasoning. Muddit (Shi et al., 2025) and MMaDA (Yang et al., 2025b) scale these ideas using shared transformers and reinforcement learning for enhanced alignment. Despite progress, unified diffusion models still face challenges in inference efficiency, sparse supervision, and architectural limitations, motivating further research in scalable, efficient multi-modal generation. Another major direction in unified multi-modal understanding and generation models adopts auto-regressive architectures. Some methods like TokLIP (Lin et al., 2025b), Harmon (Wu et al., 2025b), Chameleon (Team, 2024), Emu3 (Wang et al., 2024), etc, utilize the VQGAN-style tokenizer to compress the high-dimensional pixel space into a compact latent space and obtain the pixel-level features. In addition to overcoming the semantic limitations inherent in pixel-based encoders, OmniGen (Xiao et al., 2025), UniWorld (Lin et al., 2025a), and ILLUME (Huang et al., 2025) facilitate CLIP-like encoders to extract high-level semantic information to improve the convergence of the generation branch. Furthermore, hybrid-based methods preserve symbolic reasoning capabilities, while employing diffusion processes for image generation to enhance global consistency and visual quality. Representative works include Show-o (Xie et al., 2024) and BAGEL (Deng et al., 2025). The former typically leverages pixel-level or continuous latent representations combined with bidirectional attention to achieve cross-modal alignment, whereas hybrid encoding methods such as BAGEL (Deng et al., 2025) integrate semantic features with pixel-level latent spaces to jointly support both understanding and generative capacities.

### 2.2 END-TO-END IMAGE TRANSLATION

End-to-end image translation can be categorized into two sub-tasks based on the target modality: Text Image Translation (TIT) and In-Image Translation (IIT). TIT focuses on translating visual text in the source language into text in the target language, representing a cross-modal process between image and text.

Most existing end-to-end image translation approaches concentrate on TIT, and many representative methods have been proposed (Chen et al., 2021; Su et al., 2021; Zhu et al., 2023; Lan et al., 2023; Salesky et al., 2024; Liang et al., 2024; Zhang et al., 2025c). CLTIR (Chen et al., 2021) first proposes the instance-level translation and regards it as a cross-linguistic recognition task. PEIT (Zhu et al., 2023) proposes an end-to-end image translation framework that bridges the modality gap with pre-trained models. (Lan et al., 2023) constructs a multi-stage training framework to mitigate the error propagation of OCR and machine translation. (Liang et al., 2024) and (Zhang et al., 2025b) are TIT methods in document domain to solve the problem of dense texts in various layouts. (Wang et al., 2025b) makes a comprehensive analysis of existing MLLM for TIT task.

In contrast, IIT aims to directly replace the source language text within the image with the corresponding target language text, without generating textual output as an intermediate result. (Qian et al., 2024) merges the TIT model and text editing model for IIT task, and (Lan et al., 2024) proposes an auto-regressive model to achieve IIT tasks in synthetic images. (Tian et al., 2025b;a) collect the caption data of videos as in-image translation translation. However, the instability of the generation model limits the development of IIT in practice.

## 3 IMTBENCH

In this section, we first define the immersive muti-modal translation task. Then we describe the dataset collection method of our dataset. Finally, a comprehensive evaluation protocol is introduced.

### 3.1 PROBLEM DEFINITION

In contrast to prior methods, we propose a novel end-to-end image translation task tailored for the era of unified multi-modal models. Given an image in source language $I_{src}$, the task is conditioned on a prompt $P(\cdot)$ which specifies both the source language $l_{src}$ and target language $l_{tgt}$. The model $\mathcal{M}$, designed as a unified end-to-end multi-modal translator, produces dual-modal outputs: the translated text $T_{tgt}$ and the translated image $I_{tgt}$, which visually embeds the translated content. The overall process is formalized in Eq. (1). Importantly, and in alignment with the assumptions of prior end-to-end image translation tasks, we restrict the model from accessing the original embedded text $T_{src}$ as an explicit input. This ensures the model performs holistic cross-modal translation without relying on intermediate text recognition.

$$[I_{tgt}, T_{tgt}] = \mathcal{M}(I_{src}, P(l_{src}, l_{tgt})). \tag{1}$$

### 3.2 DATA CURATION

To solve the immersive muti-modal translation problem, we first construct a comprehensive dataset, IMTBench, which is constructed through three complementary data collection pipelines, each targeting different sources and modalities to ensure both diversity and quality. The detailed process is introduced in Appendix C.

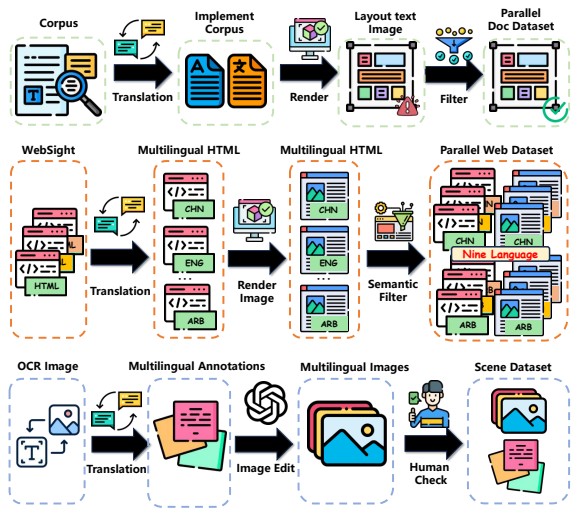

**Document.** We begin with large-scale parallel textual corpora cross language, which are implemented into nine target languages. These translated documents are then implemented into structured layouts and rendered into image form. To guarantee dataset reliability, we apply filtering procedures to remove low-quality or noisy samples. The resulting Parallel Document Dataset contains well-aligned multilingual text–image pairs suitable for training cross-lingual multimodal models.

Figure 2: The curation description of IMTBench. From top to bottom: (1) *Document* focuses on multilingual document translation with structured layouts, (2) *Web* targets text rendering and fidelity in webpage-style images, and (3) *Scene* emphasizes instruction-driven editing of scene text in natural images.

**Web.** The second pipeline leverages multilingual web resources. Starting from raw HTML pages collected via WebSight (Laurençon et al., 2024), we perform automatic translation into several target languages. The translated HTML content is rendered into corresponding multilingual images, followed by semantic filtering to ensure alignment across languages. This process yields a large-scale Parallel Web Dataset spanning nine languages, capturing real-world webpage structures and multilingual contexts.

**Scene.** The third pipeline focuses on real-world images containing textual content. Optical Character Recognition (OCR) is applied to extract the embedded text, which is then translated into multiple languages. The translated annotations are reintegrated into the images via editing, generating multilingual variants of the original image. Human verification ensures accuracy and naturalness. This procedure produces a Real-World Multilingual Image Dataset with high fidelity to authentic visual environments.

### 3.3 EVALUATION PROTOCOLS

In this section, we introduce the key evaluation metrics used to assess both the textual and visual quality of our system's outputs. We employ COMET (Rei et al., 2020) for translation quality, OCR

accuracy to measure text fidelity, and a masked variant of LPIPS (Zhang et al., 2018) to evaluate the perceptual consistency of edited images, focusing on background preservation. Accordingly, we denote the three metrics as $S_{text}$, $S_{align}$ and $S_{vision}$ based on their respective modalities.

**COMET.** To evaluate the quality of machine translation outputs, we employ Crosslingual Optimized Metric for Evaluation of Translation (COMET) (Rei et al., 2020) as one of our primary evaluation metrics. COMET is a neural-based metric that leverages multilingual pre-trained language models and is fine-tuned on human-annotated data. Formally, given a source sentence $T_{src}$, a reference translation $T_{tgt}$, and a candidate translation $\hat{T}_{tgt}$, COMET computes a quality score $S_{text}$ as Eq. (2), where $f_\theta$ denotes the neural scoring model, which outputs a scalar value representing the predicted translation quality.

$$S_{text} = f_\theta(T_{src}, T_{tgt}, \hat{T}_{tgt}). \tag{2}$$

**OCR Accuracy.** We evaluate text editing performance using the OCR score $S_{align}$, based on word-level normalized edit distance with optimal alignment. Given the target text $T_{tgt}$ and the OCR-recognized prediction $M_{ocr}(\hat{I}_{pred})$, we segment them into word sequences $G = \{g_i\}_{i=1}^n$ and $P = \{p_j\}_{j=1}^m$. A cost matrix of normalized edit distances $C_{ij} = \frac{E(g_i,p_j)}{\max(|g_i|,|p_j|)}$ is constructed, and the best matching is obtained to compute as Eq. (3). This metric captures word-level accuracy between predicted and target texts across languages.

$$S_{align} = 1 - \frac{1}{K} \sum_{(i,j) \in \Pi} \frac{E(g_i, p_j)}{\max(|g_i|, |p_j|)}, \quad K = \min(n, m). \tag{3}$$

**Mask LPIPS.** To better evaluate the perceptual quality of edited images, we adopt the Learned Perceptual Image Patch Similarity (LPIPS) metric (Zhang et al., 2018), which measures perceptual distances in deep feature space. Given a binary mask $M \in \{0,1\}^{H \times W}$, where $M_{hw} = 1$ indicates the target background region and $M_{hw} = 0$ corresponds to the edited textual or foreground region, we modify the LPIPS calculation as Eq. (4). This formulation ensures a more faithful evaluation of whether the background consistency is preserved during the editing process, while ignoring the intended modifications inside the edited text areas. To facilitate consistent comparison across settings, we normalize $S_{vision}$ vision using a $1-$ transformation.

$$S_{vision} = 1 - \sum_l \frac{1}{\sum_{h,w} M_{hw}} M_{hw} \omega_l ||\phi_l(I_{tgt} - \phi_l(\hat{I}_{tgt})||_2^2. \tag{4}$$

At last, we propose the aggregated score $S$, calculated by above three protocols. $S$ is defined the mean value of three normalized sub-metrics $S = \frac{1}{3}(S_{text} + S_{align} + S_{vision})$.

## 4 EMPIRICAL EXPERIMENTS

Following the construction of IMTBench, we systematically evaluated a diverse set of models, including representative commercial cascaded APIs (Tencent[2] and Youdao[3]), proprietary unified multimodal generation and understanding models (Seedream and GPT-4o), and open-source unified generation and understanding models (Qwen-Image, Janus-Pro, Bagel, and Uniworld). Empirical analyses were conducted across varying model architectures (Section 4.1), application scenarios (Section 4.2, and input–output languages (Section 4.3). All experiments employed the official pre-trained weights and inference scripts, ensuring reproducibility, with detailed configurations provided in the Appendix.

### 4.1 PERFORMANCES ON DIFFERENT PARADIGMS

Table 1 presents the immersive muti-modal translation performance of representative methods under different paradigms. Commercial multi-stage pipeline methods achieve the highest $S_{text}$ and $S_{align}$, while maintaining the lowest $S_{vision}$ across most scenarios. The multi-stage pipeline architecture,

---

[2] tmt.tencentcloudapi.com
[3] https://openapi.youdao.com/ocrtransapi

Table 1: Immersive muti-modal translation performances of representative methods in different paradigms. All reported values in the table are percentages. $S_{avg}$ indicates the average value of aggregated score $S$. **Bold numbers denote the best performance in each column.**

| Methods | Document | | | Web | | | Scene | | | $S_{avg}$ |
|---|---|---|---|---|---|---|---|---|---|---|
| | $S_{text}$ | $S_{align}$ | $S_{vision}$ | $S_{text}$ | $S_{align}$ | $S_{vision}$ | $S_{text}$ | $S_{align}$ | $S_{vision}$ | |
| *Commercial Multi-stage Pipeline* | | | | | | | | | | |
| Tencent Translation | **64.3** | **79.0** | **88.2** | 77.2 | **75.8** | **86.4** | 61.6 | 55.2 | 38.0 | **73.1** |
| Youdao Translation | 60.8 | 77.8 | 87.6 | 73.1 | 75.8 | 85.5 | 64.3 | **59.0** | **45.5** | 72.7 |
| *Proprietary Unified Multi-modal Model* | | | | | | | | | | |
| Seedream3.0 (Gao et al., 2025) | 46.2 | 35.7 | 81.4 | 66.3 | 26.5 | 78.6 | 39.5 | 5.0 | 52.6 | 51.1 |
| GPT-4o (OpenAI, 2024) | 56.9 | 27.2 | 57.8 | **77.5** | 26.3 | 75.0 | **68.6** | 12.5 | 51.8 | 51.6 |
| *Open-source Unified Multi-modal Model* | | | | | | | | | | |
| Qwen-Image (Wu et al., 2025a) | 49.0 | 5.7 | 48.8 | 66.6 | 7.4 | 82.8 | 44.2 | 2.0 | 47.2 | 40.9 |
| Janus-Pro (Chen et al., 2025c) | 30.3 | 1.0 | 45.0 | 20.3 | 0.6 | 50.5 | 31.2 | 0.1 | 49.2 | 25.1 |
| Bagel (Deng et al., 2025) | 31.0 | 1.9 | 72.6 | 31.0 | 3.0 | 84.1 | 31.6 | 0.4 | 50.9 | 35.3 |
| UniWorld (Lin et al., 2025a) | 48.3 | 7.2 | 65.5 | 59.4 | 7.1 | 78.6 | 44.4 | 2.7 | 40.1 | 41.3 |

typically combining dedicated OCR modules and mature machine translation systems, benefits from specialized component optimization.

Proprietary unified multi-modal models in this study cannot generate text and images simultaneously. To approximate unified generation and understanding, we evaluated them by pairing their respective generation APIs (GPT-Image-1 and Seedream-3.0) with corresponding understanding APIs (GPT-4o and Doubao1.6). For understanding-focused tasks such as machine translation, they achieve performance comparable to commercial multi-stage pipelines. However, significant misalignment between generated content and images was observed, likely due to multi-step inference. On the $S_{vision}$ metric, Seedream excels in simple-background scenarios, indicating strong background adherence, whereas GPT tends to excessively modify original images, reflecting limited preservation of visual fidelity.

The final part of Table 1 indicates that open-source generation and understanding models still have substantial room for improvement. This may be attributed to resource limitations, which have prevented these models from being trained on proprietary tasks or multilingual datasets. Notably, Qwen-Image and UniWorld, which are based on Qwen2.5VL-7B, demonstrate relatively strong performance on translation tasks. However, their performance on the $S_{align}$ metric, reflecting text-editing ability, and the $S_{vision}$ metric, reflecting instruction-following capability, remains considerably lower than that of the previously discussed pipelines. Furthermore, JanusPro and Bagel, which employ more lightweight architectures, exhibit significantly lower generation and understanding scores across all metrics. These Finding suggest that the current unified fine-tuning strategies for generation and understanding modules may not effectively promote true synergy between content generation and comprehension.

> ***Finding 1:*** Both open-source and closed-source generative–understanding models exhibit a considerable performance gap compared with existing commercial cascaded pipelines on the IMT task, suggesting that unified multimodal models still have substantial room for improvement in coordinating understanding and generation.

## 4.2 Performances on different scenarios

Unified generation and understanding models also exhibit intriguing patterns across different scenarios. For multi-stage pipeline approaches, *Document* and *Web* scenarios are relatively simple, and using text erasure combined with manual rendering has little impact on the *Scene* images, resulting in consistently low $S_{vision}$ scores. In contrast, real-world settings, which involve complex factors such as natural lighting, occlusions, and authentic noise, impede the performance of multi-stage pipelines. Although unified generation and understanding models cannot generalize well to IMT tasks, their exposure to large-scale image generation and editing data endows them with a strong ability to preserve image naturalness, highlighting their substantial potential—particularly in complex, real-world scenarios. For example, compared with the *Document* and *Web* subsets, the $S_{vision}$ gap is notably reduced in the Scene subset, indicating improved alignment under more complex visual conditions.

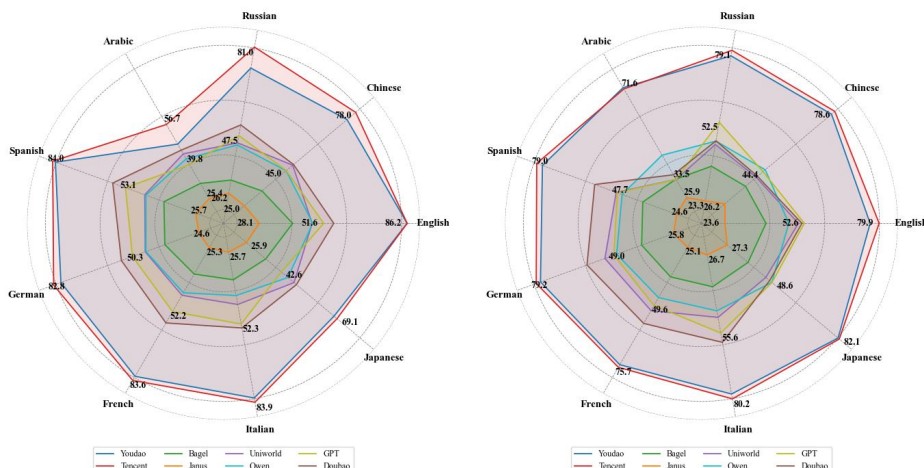

Figure 3: Performance comparison of different immersive translation solutions across multiple languages. Results on the left panel show performance when varying the target language, while results on the right panel illustrate performance when varying the source output language. We show the number label of Tencent, GPT-Image, and Janus.

> ***Finding 2:*** Unified generation and understanding models demonstrate strong potential in preserving image naturalness, particularly in complex real-world scenarios, yet their performance on multimodal translation and instruction-following tasks remains limited, highlighting the need for improved strategies to effectively coordinate content generation and comprehension.

## 4.3 PERFORMANCES ON DIFFERENT LANGUAGES

In the IMT setting, multilinguality poses an additional challenge beyond perception–understanding synergy, as unified multi-modal models are expected to operate robustly across diverse linguistic contexts. We evaluate performance under varying source $l_{src}$ and target $l_{tgt}$ languages using an aggregate metric $S$.

As shown in Fig. 3, while Latin languages (English, French, German, Spanish, Italian), Cyrillic (Russian), and Chinese are relatively well supported, Arabic and Japanese exhibit significant performance drops. This gap is largely attributable to data scarcity and script-specific challenges: Japanese, despite its partial overlap with Chinese characters, lacks sufficient training resources to generalize effectively; Arabic further suffers from limited annotated corpora, and its unique orthography and right-to-left writing system exacerbate difficulties in both understanding and generation.

To complement the above analysis on target languages, we further examine the impact of different source languages. As shown in the right panel of Fig. 3, the overall conclusions remain consistent with the target-language evaluation; however, the performance gaps across source languages are notably smaller. This suggests that the primary cross-lingual disparity arises at the output level, whereas the input side exerts comparatively limited influence.

> ***Finding 3:*** These Finding highlight the uneven cross-lingual generalization of current models, and underscore that unified generation understanding models, while effective in high-resource languages, require more balanced multilingual resources and tailored design to handle low-resource, non-Latin scenarios.

## 4.4 VISUALIZATION

Figure 4 presents qualitative results from IMTBench across three scenarios and multiple languages, comparing representative multimodal models. Due to space constraints, we include the Tencent API as a commercial cascade-based translation system, SeedEdit and GPT-Image as closed-source unified generation and understanding models, and Qwen-Image as an open-source counterpart. The visualizations are consistent with the quantitative analysis in Table 1. Specifically, the Tencent API performs strongly in document and webpage scenarios, but suffers in real-world settings where ren-

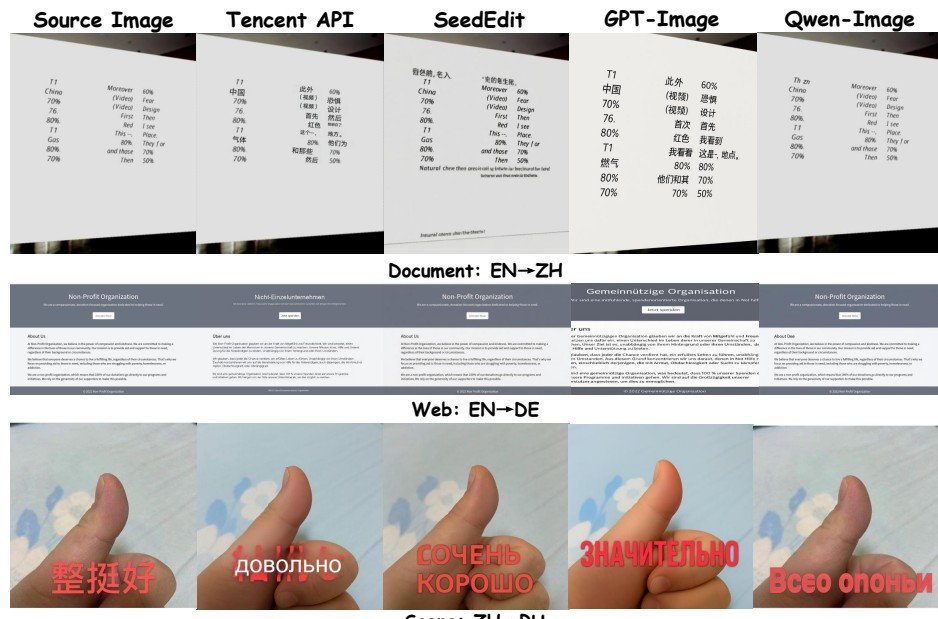

Figure 4: Visualization of unified multi-modal models with different architectures on the IMT task.

Table 2: Comparison of UniWorld model performance before and after fine-tuning on the *Scene* benchmark. All reported values in the table are percentages.

| Settings | $S_{text}$ | $S_{align}$ | $S_{vision}$ |
|---|---|---|---|
| UniWorld | 44.4 | 2.7 | 40.1 |
| + Fine-tuning | 57.5 $_{(+13.1)}$ | 13.8 $_{(+11.1)}$ | 47.8 $_{(+7.7)}$ |

dered text often appears misaligned with the background due to the limitations of its cascade design. SeedEdit and Qwen-Image exhibit limited IMT capabilities in text-dense scenes, yet achieve more coherent results in real-world cases, indicating their potential for this task. GPT-Image demonstrates the strongest overall ability, successfully handling translation across all three scenarios and producing visually harmonious outputs, but tends to over-modify the original content, particularly in background adherence during editing tasks.

## 5 MORE FINE-TUNING EXPLORATION

Furthermore, we fine-tune open-source training scripts of unified generation and understanding models, including Janus-Pro (Chen et al., 2025c;b), UniWorld (Lin et al., 2025a; Wang et al., 2025c), and Bagel (Deng et al., 2025). Following the data collection pipeline described in Section 3.2, we construct approximately 1M parallel images across nine languages, termed as IMT-1M, with the majority drawn from Document and Web scenarios, and additional Scene samples curated outside IMTBench. For fairness, we adopt the training configurations from the original papers, and discuss results separately for both the convergence behavior and the generation branch variants.

### 5.1 CONVERGENCE

Fig. 5 presents the loss curves of three representative models trained on IMT-1M until convergence, revealing markedly different optimization behaviors for the IMT task. Bagel, which jointly optimizes generation and understanding, exhibits a rapid initial loss decrease followed by a slower convergence, reaching stability around 40k steps. UniWorld, leveraging the pre-trained Qwen2.5VL (Bai et al., 2025) and FLUX-KonText model (Labs et al., 2025), starts from a substantially lower loss due to strong pre-training and experiences oscillatory decay over the subsequent 35k steps. In contrast, Janus, representing a purely autoregressive unified model, converges more slowly; around 10k steps it briefly becomes trapped in a local optimum before gradually decreasing. These observations indicate that in unified generation and understanding models, purely autoregres-

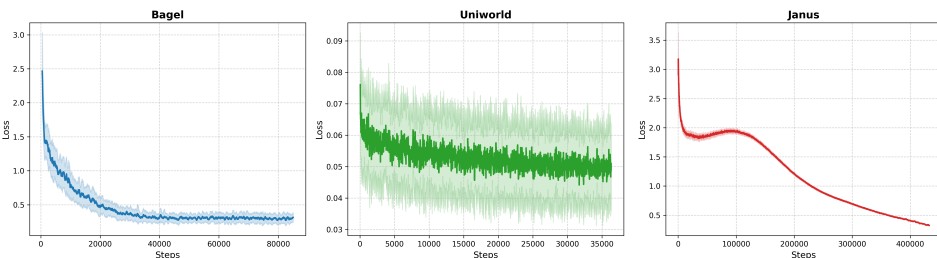

Figure 5: Loss of Finetuning on IMT-1M.

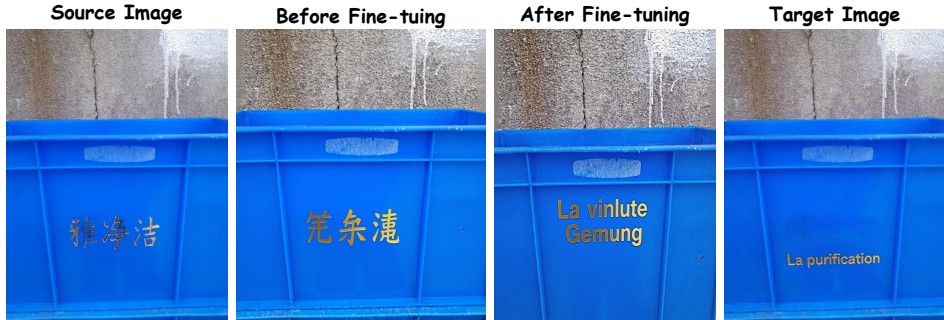

Figure 6: Visualization of Finetuning on IMT-1M. The prompt is "*Translate Chinese into French.*"

sive architectures are limited by slower convergence, whereas strong diffusion-based pre-training enables faster generalization on the IMT task.

> **Finding 4:** *A strong foundation of pre-trained understanding and generation components is a critical prerequisite for effective synergy during fine-tuning on the IMT task.*

## 5.2 RESULTS

Based on the above Finding, we report the performance of the UniWorld model before and after fine-tuning, focusing on the variant that achieved the best convergence. To simplify the experimental setup, we evaluate on the *Scene* subset. As shown in Table 2, fine-tuning brings substantial improvements across all three metrics; however, there remains a notable gap compared to the multi-stage expert pipeline in Table 1. This suggests that fine-tuning the DiT alone, while effective in boosting performance, is insufficient for fully aligning the generation and understanding components. We further visualize model outputs before and after fine-tuning and observe a clear progression. As shown in Fig. 6, the unified generation and understanding model initially lacks immersive multimodal translation capability, then gradually learns target-language glyph information, and eventually acquires correct semantic knowledge to produce accurate translations. These insights may inspire future efforts toward developing unified multi-modal models with stronger synergy between generation and understanding.

> **Finding 5:** *The unified generation-understanding model learns in a progressive, hierarchical manner, first acquiring glyph and shape information before mastering the semantic knowledge required for accurate translation.*

## 6 CONCLUSION

In this work, we propose Immersive Multimodal Translation (IMT) as a novel proxy task to evaluate unified multimodal generation and understanding models. We introduce IMTBench, a systematic benchmark spanning diverse scenarios and languages, and conduct extensive evaluations and fine-tuning to reveal structural and performance gaps across models. We believe this task can inspire the community to enhance generation and understanding synergy and guide targeted optimization of unified multimodal models.

REPRODUCIBILITY STATEMENT

We have made extensive efforts to ensure that the results reported in this work are reproducible. All model architectures, training procedures, and hyperparameter settings are described in the main text (Sections 3) and detailed further in the Appendix (Appendix C–E). For the datasets used in our experiments, we provide complete descriptions of preprocessing and filtering steps in the supplementary materials. All evaluation metrics are formally defined in Section 3.3, enabling consistent replication of our analysis. Additionally, the source code and scripts used for training, inference, and evaluation will be made publicly available as anonymized supplementary material, facilitating direct reproduction of the reported results. Readers are referred to these resources for all necessary details to reproduce the experiments and analyses presented in this work.

ETHICS STATEMENT

All authors have read and adhered to the ICLR Code of Ethics. This work focuses on constructing a proxy task for evaluating unified multi-modal generation and understanding models. We use objective evaluation protocols, which do not involve direct experimentation on human subjects. All datasets used are either publicly available or used under appropriate licenses, and any personal information has been anonymized to protect privacy. We are aware of potential societal impacts of multimodal AI systems, including misuse for generating misleading content or biased outputs. In our experiments, we take care to evaluate model behavior across diverse languages and scenarios to mitigate unintended bias. No datasets or methods used are expected to cause harm to individuals or communities. We encourage responsible use and recommend that future users of the proposed models follow relevant legal, privacy, and fairness guidelines. Any conflicts of interest have been disclosed, and all research practices adhere to established standards of scientific integrity.

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

The appendix includes the following aspects:

- **A**: Use of Large Language Models
- **B**: Comparisons of different machine translation benchmarks.
- **C**: Details of IMTBench curation.
- **D**: Details of Experiment Settings.

## A  USE OF LARGE LANGUAGE MODELS

In this work, large language models (LLMs) are used solely as generally purpose assistive tools to improve the clarity, grammar, and readability of the manuscript. LLMs are not used for research ideation, data analysis, model development, or any other scientific decision-making. All scientific content, ideas, results, and conclusions presented in this paper are independently produced by the authors. The authors take full responsibility for the accuracy and integrity of the work, including any content that was refined or edited with the assistance of LLMs. No information generated by LLMs that could constitute plagiarism, fabrication, or scientific misconduct has been included.

## B  COMPARISONS OF DIFFERENT MACHINE TRANSLATION BENCHMARKS.

As an extension of multimodal machine translation, Immersive Multi-modal Translation (IMT) requires the joint construction of image–text inputs and outputs, making dataset creation more challenging than in Text Image Translation (TIT) and In-Image Translation (IIT). As summarized in Table 3, our dataset is competitive in scale and uniquely characterized by multilingual parallelism, cross-modal input–output, and real-world scenarios. Multilingual parallelism enhances data efficiency, cross-modal input–output enables the assessment of generation and understanding synergy in unified multimodal models, and real-world data provides conditions for practical applications. Moreover, the cross-modal setting can also provide additional data support for TIT and IIT tasks.

Table 3: The comparison between multi-modal translation dataset. $^\star$ indicates the original paper reports the instance number, rather than the number of images.

| Dataset | Train | Eval | Languages | Parallel | Modality | Real Scene |
|---|---|---|---|---|---|---|
| *TIT Datasets* | | | | | | |
| OCRMT-30K (Zhu et al., 2023) | 30k | 1.2k | 2 | ✗ | Text-Only | ✓ |
| MTIT6 (Qian et al., 2024) | - | 6k | 4 | ✗ | Text-Only | ✓ |
| AibTrans (Wang et al., 2025b) | - | 7k | 8 | ✓ | Text-Only | ✓ |
| MIT-10M $^\star$ (Li et al., 2025) | 10M | 10.4k | 14 | ✓ | Text-Only | ✓ |
| *IIT Datasets* | | | | | | |
| Translatotron-V (Lan et al., 2024) | 81.7k | 3.5k | 4 | ✓ | Image-Only | ✗ |
| DebackX (Tian et al., 2025b) | 75k | 8.2k | 2 | ✗ | Image-Only | ✗ |
| PRIM (Tian et al., 2025a) | 6.8M | 17k | 6 | ✗ | Image-Only | ✓ |
| *IMT Datasets* | | | | | | |
| IMTBench (Ours) | 1M | 2k | 9 | ✓ | Image-Text Pair | ✓ |

## C  DETAILS OF IMTBENCH CURATION.

### C.1  DATA COLLECTION

While Section 3.2 offers a concise overview of the IMT data construction process owing to space limitations, this section provides a comprehensive account with sufficient details to guarantee reproducibility.

**Document.** In the Document subset, we employ the SynthDoG engine to simulate rich-text document images resembling real-world scenarios. We first collect parallel corpora[4], using subtitle files

---

[4]https://github.com/ajinkyakulkarni14/TED-Multilingual-Parallel-Corpus

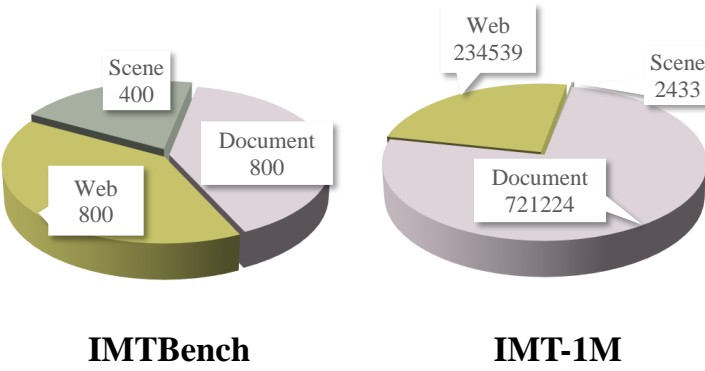

Figure 7: Visualization of the data distribution of IMTBench and IMT-1M across the three scenarios, illustrating the relative proportions of samples in *Document*, *Web*, and *Scene* settings.

from TED talks as the primary source. Although these subtitles are multilingual and roughly aligned, inconsistencies in word order across languages make direct utilization infeasible. To address this, we leverage a lightweight translation expert model[5] (0.6B parameters) to complete the parallel corpus efficiently at scale, followed by SynthDoG (Kim et al., 2022) rendering to generate structured document images. To ensure translation quality, we further apply automatic filtering with Qwen3-8B (Yang et al., 2025a). For IMTBench construction, we select 100 nine-way parallel samples that cover diverse content, and randomly assign one language as the source, yielding 800 test cases in total. For IMT-1M, we generate around 80k parallel samples (720k images), as shown in Fig. 7.

**Web.** In the Web subset, we build upon WebSight v2 (Laurençon et al., 2024), a synthetic dataset containing 2 million pairs of HTML code and corresponding screenshots. Compared to WebSight v1, this version explicitly encodes the placement of illustrations, better reflecting realistic webpage layouts. However, most illustrations are invalid URL placeholders. To address this, we collected icon images from the public web and adaptively scaled them according to the resolution specified in the original URLs, thereby preserving the original page structure. For translation, we adopt the same lightweight expert model used in the Document subset. We further crawl over 30k raw webpages and render them with Selenium. Since Selenium-based rendering can produce misalignments between text and screenshots, we apply Qwen2.5VL-7B for automatic filtering. As a result, we obtain a parallel dataset of over 20k webpages, comprising more than 234k aligned text–image pairs.

**Scene.** Compared with the Document and Web subsets, the Scene subset lacks a stable data construction engine that can perform large-scale editing and translation of scene images. To address this limitation, we adopt an integrated strategy to construct real-scene data. We first collect a set of real-world images from OCR datasets, which contain precise OCR annotations. Based on these annotations, we build parallel text labels in nine languages. Next, for each pair of original and translated text, we provide the inputs to two editing models with strong text-editing capabilities, namely GPT-Image and SeedEdit. Unlike the evaluation setting, the prompts here explicitly include both the source and translated text to reduce the difficulty of model comprehension. In practice, we find that SeedEdit adheres more faithfully to the original image, but performs poorly in Japanese, Russian, and Arabic. Therefore, we adopt SeedEdit outputs for Latin languages and supplement the three challenging languages with GPT-Image results. All generated images are further manually verified, retaining only those with natural and correct rendering. This process results in 2,833 paired samples, from which we randomly select 400 for IMTBench, while the remaining are incorporated into IMT-1M to enhance the realism of the training set.

## C.2 DATA STATISTICS

Our IMTBench comprises multilingual multi-modal translation samples covering nine languages. To illustrate the data characteristics, we provide three complementary visualizations. Figure Fig. 9

---

[5] https://huggingface.co/facebook/nllb-200-distilled-600M

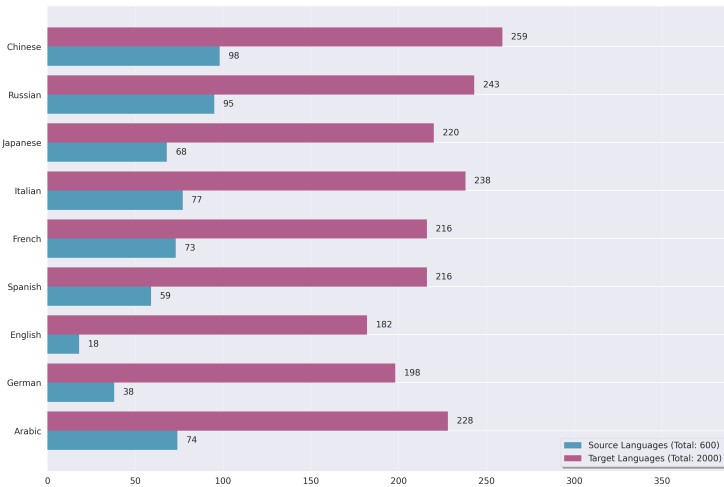

Figure 8: Data distribution across nine languages in IMTBench. Due to data organization constraints, the benchmark contains 600 reference images and 2,000 target images, yielding 2,000 test cases.

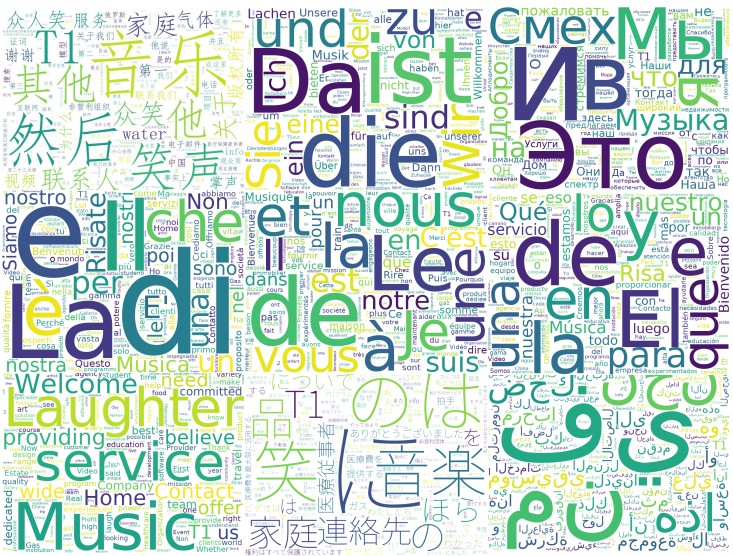

Figure 9: Word clouds showcasing top questions in various languages.

presents word clouds highlighting the most frequent tokens across different languages, reflecting the vocabulary diversity. Fig. 8 shows the frequency distribution of each language as source and target, demonstrating the balance between input and output directions. Figure Fig. 10 further reports the token length distribution of both source and target texts, where tokenization is performed using the Qwen2.5VL-7B tokenizer. In conclusion, these statistics provide a comprehensive view of the dataset composition and linguistic variation.

### C.3 VISUALIZATION OF IMTBENCH.

Fig. 11 illustrates the parallel visualization of IMTBench in nine languages. By leveraging a limited number of images, this data construction approach scales to a vast number of translation pairs, significantly enhancing the efficiency of data utilization. Fig. 12 shows the annotation form, corresponding to the second column of Fig. 11.

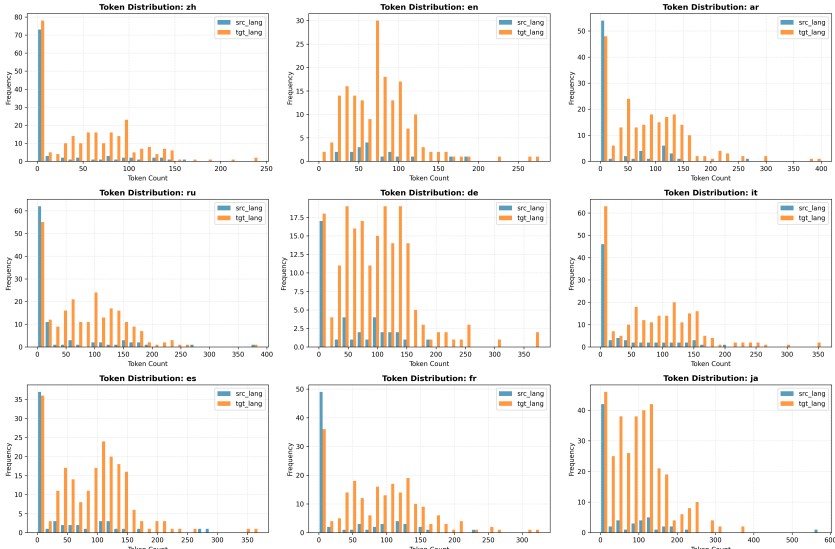

Figure 10: Token length distribution across nine languages in IMTBench. Due to data organization constraints, the benchmark contains 600 reference images and 2,000 target images, yielding 2,000 test cases.

## D  DETAILS OF EXPERIMENT SETTINGS.

### D.1  INFERENCE SETTINGS

For models that cannot generate both text and images simultaneously, we employed a comprehension model from the same developer to produce the corresponding text–image translations. The details of settings are illustrated by Table 4. **We argue that this setup, using a paired understanding model, can effectively simulate the inference behavior of unified generation and understanding models.** For the experiments reported in Table 1, we used a minimal prompt format, as follows:

*"Translate all texts in this image from $\{l_{src}\}$ to $\{l_{tgt}\}$, and replace all texts with translated texts."*

Table 4: Practical implementation of evaluation experiments for unified generation and understanding models.

| Methods | Implementation |
|---|---|
| GPT-4o | GPT-4o + GPT-Image-1 |
| Seedream3.0 | Doubao1.6 + SeedEdit3.0 |
| Qwen-Image | Qwen2.5VL-7B + Qwen-Image-Edit |

For Janus-Pro and UniWorld, we inference on IMTBench with the edit-version Shared-GPT4o-Image (Chen et al., 2025b) and GPT-Edit-1.5M (Wang et al., 2025c), which has editing capability of general image editing. The detailed hyper-parameters follows official settings.

### D.2  FINE-TUNING SETTINGS

To improve the performance of open-source generation–understanding models on the IMT task, we fine-tuned three models—Bagel, Janus-Pro, and UniWorld—on the IMT-1M dataset. The pre-trained checkpoints used were Bagel-MoT-7B, Janus-Pro-7B, and GPT-Image-Edit (with FLUX-Kontext). All experiments were conducted on 8 NVIDIA H100 GPUs.

For Janus-Pro, key training settings included 1 epoch, a per-GPU batch size of 2, gradient accumulation over 8 steps, and a learning rate of 5e-5. Training was launched with 8 processes on a single machine using the standard DeepSpeed multinode launcher.

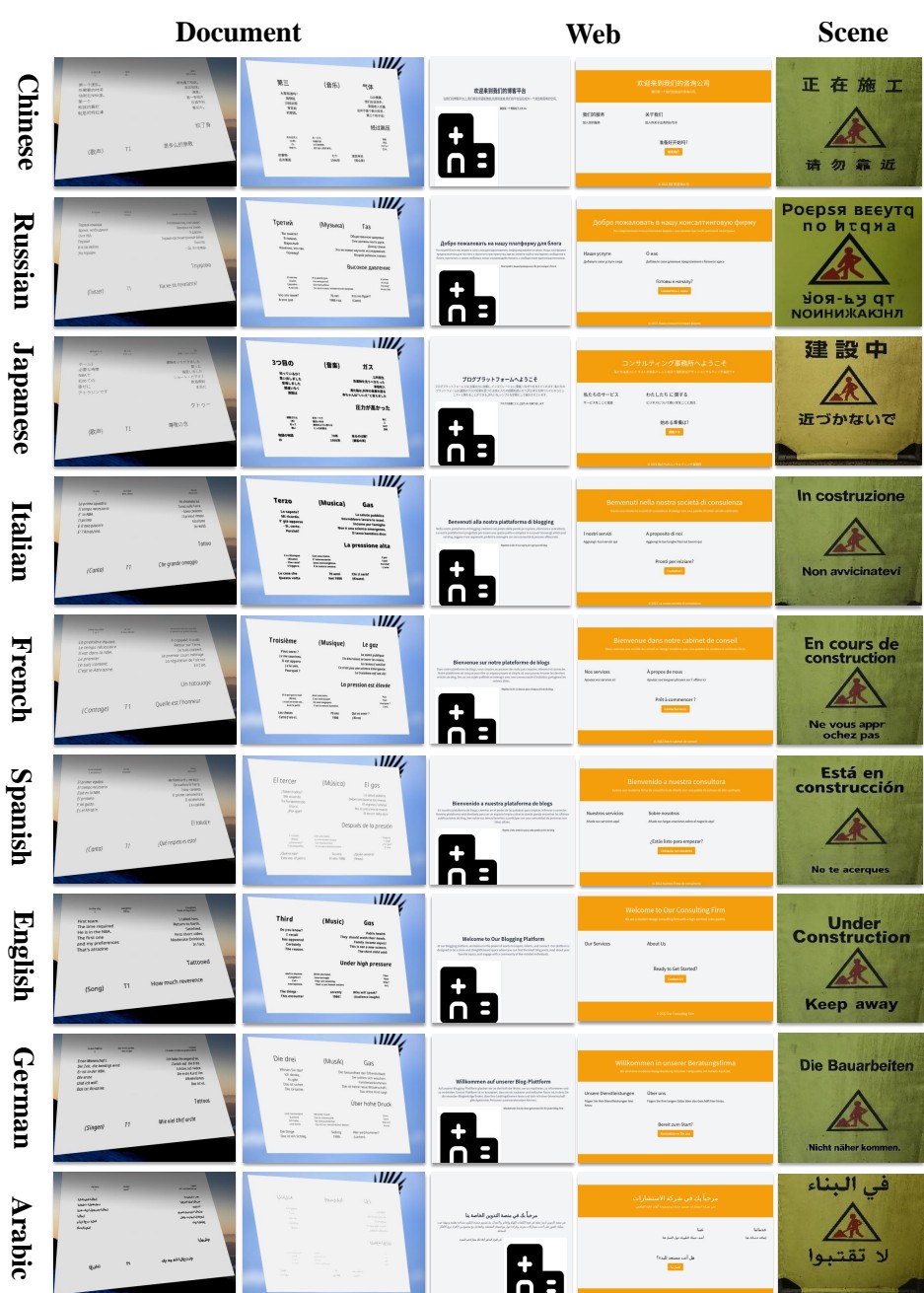

Figure 11: Visualization of IMTBench.

For UniWorld, we followed a similar setup with 1 training epoch, a batch size of 1, gradient accumulation of 16 steps, and a learning rate of 1e-5 using the AdamW optimizer. Mixed precision (bf16) and gradient checkpointing were enabled, and the model was fine-tuned on the textual and visual branches jointly.

For Bagel, fine-tuning was performed from the pre-trained Bagel-MoT-7B checkpoint with a maximum latent size of 64, learning rate of 2e-5, automatic checkpoint resume, and a per-GPU batch size of 1.

```
{
    "source_lang": "ru",
    "source_text": "Третий (Музыка) Газ Вы знаете? Я помню. Взрослый Конечно, это так. Почему? Общественное здоровье Они должны мыть руки. Доход семьи Это не новое научное исследование. Второй ребенок сказал: Высокое давление И никому (Смех) Сделайте… И легкие. И в одной руке. Что интересно? Они волшебны. Это всего лишь человеческая природа. И затем И тогда Почему? Искусство Что это такое? В этот раз 70 лет 1986 год Кто это будет? (Смех)",
    "source_image_path": "ru_image_1715.jpg",
    "all_languages": {
        "en": {
            "text": "Third (Music) Gas Do you know? I recall has appeared Certainly The reason. Public health They should wash their hands. Family income aspect This is not a new science. The third child said: Under high pressure and to anyone (Laughter) Cut--
 And leptons. With one hand, Interestingly They are amazing. That's just human nature. Then Then Why? Art, The things - This encounter seventy 1986年 Who will speak? (Audience laughs)",
            "image_path": "en_image_1715.jpg"
        },
        "zh": {
            "text": "第三 （音乐）气体 大家知道吗？ 我想起 已经出现 肯定会 的原因。 公众健康，他们应该洗手。 家庭收入方面 这并不是个新兴科学， 第三个孩子说： 经过高压 并向任何人 (众笑) 切-- 和轻子。 用一只手， 有趣的是 它们很神奇。 那只是人类的本性。 接着 然后 为什么？ 艺术、的事物- 这次邂逅 七十 1986年 谁会来说 （观众笑）",
            "image_path": "zh_image_1715.jpg"
        },
        "ar": {
            "text": "الثالثة (موسيقى) الغاز هل تعلمون؟ أتذكر لقد ظهر بالتأكيد السبب صحة العامة يجب أن يغسلوا يدهم الدخل العائلي هذا ليس علماً حديثاً. وقال الطفل الثالث: الضغط العالي و لم يُعطى أي شخص (ضحك) - تقطّع و خفيفة بيد واحدة المثير للاهتمام إنها رائعة هذا هو طبيعة الإنسان ثم ثم لماذا؟ الفن \"أشياء\" هذه المرة 70 عام 1986 من سيأتي (ضحك)",
            "image_path": "ar_image_1715.jpg"
        },
        "de": {
            "text": "Die drei (Musik) Gas Wissen Sie das? Ich denke, Es gibt Das ist sicher. Die Ursache. Die Gesundheit der Öffentlichkeit Sie sollten sich waschen. Familieneinkommen Das ist keine neue Wissenschaft. Das dritte Kind sagt: Über hohe Druck Und niemandem (Lachen) Ich habe… Und leicht. mit einer Hand, Das ist interessant. Sie sind wunderbar. Das ist nur menschliches Wesen. Dann Dann Warum? Kunst, Die Dinge Das ist ein Schlag. Siebzig 1986: Wer wird kommen? (Lachen)",
            "image_path": "de_image_1715.jpg"
        },
        "it": {
            "text": "Terzo (Musica) Gas Lo sapete? Mi ricordo. E' già apparso -
 Sì, certo. Perché? La salute pubblica Dovrebbero lavare le mani. Income per famiglie Non è una scienza emergente. Il terzo bambino dice: La pressione alta E a chiunque (Risate) -
 Che cosa? E leggero. Con una mano, E' interessante Sono meravigliose. È la natura umana. E poi E poi Perché? L'arte Le cose che Questa volta 70 anni Nel 1986 Chi ci sarà? (Risate)",
            "image_path": "it_image_1715.jpg"
        },
        "es": {
            "text": "El tercer (Música) El gas ¿Saben todos? Me acuerdo. Ya ha aparecido Eso sí. ¿Por qué? La salud pública, Deberían lavarse las manos. El ingreso familiar No es una ciencia nueva. El tercer niño dijo: Después de la presión Y a nadie (Risas) -
 ¿Qué es eso? Y las pequeñas. Con una mano, Lo interesante es que Son maravillosos. Es la naturaleza humana. Después Y luego ¿Por qué? El arte ¿Qué es eso? Esta vez, el perro. Se veía El año 1986 ¿Quién vendrá? (Risas)",
            "image_path": "es_image_1715.jpg"
        },
        "fr": {
            "text": "Troisième (Musique) Le gaz Vous savez ? Je me souviens. Il est apparu Je le sais. Pourquoi ? La santé publique Ils devraient se laver les mains. Le revenu familial Ce n'est pas une science émergente. Le troisième enfant dit: La pression est élevée Et à qui que ce soit (Rires) Je suis en train de… Avec le petit. Avec une main, C'est intéressant Ils sont magiques. C'est la nature humaine. Puis Puis Pourquoi ? L'art, Les choses Cette fois-ci, 70 ans 1986 Qui va venir ? (Rires)",
            "image_path": "fr_image_1715.jpg"
        },
        "jp": {
            "text": "3つ目の (音楽) ガス 知っているか? 思い出しました 登場しました 間違いなく 原因は 公共衛生 洗濯物を洗うべきだった 家庭収入 進化論は,科学の発展を図る 赤ちゃんは\"いいえ\"と答えました 圧力が高かった 保護された (笑) 切って 軽い 手を一つで 面白いのは 魔法のように見える ヒトの本質は 後に に なぜ? 芸術 物語の物語 の 70年 1986年 来るのは誰? (観客の笑)",
            "image_path": "jp_image_1715.jpg"
        }
    }
}
```

Figure 12: Annotations of IMTBench.

This setup ensured a consistent and comparable training protocol across all open-source models while adapting them to the IMT task.

