# OpenReview forum: "Immersive Multimodal Translation: A Proxy Task for Cross-modal and Objective Evaluation of Unified Models"
_ICLR.cc/2026/Conference — Submitted to ICLR 2026_

### Official Review · Reviewer_64m2 · 2025-10-19

**Soundness:** 2
**Presentation:** 3
**Contribution:** 2
**Rating:** 2
**Confidence:** 4

**Summary:**

This paper proposes Immersive Multimodal Translation (IMT), which translates text in images while keeping visual context . The authors further propose IMTBench, a benchmark with 3 scenarios, 9 languages, and 2k samples. The authors use three metrics (translation quality, background fidelity, text rendering) to evaluate models. Experiments show open-source/closed-source models still lag behind commercial systems, but fine-tuning helps.

**Strengths:**

1. The paper is generally well-written and easy to follow.
2. The three-dimensional evaluation metrics (text, alignment, vision) are clear and comprehensive, avoiding one-sided assessments.
3. Fine-tuning experiments on IMT-1M provide practical insights (e.g., model convergence differences), which help improve unified models.

**Weaknesses:**

1. My biggest concern about this paper is that The IMT task may not truly assess the core multi-modal synergy of unified understanding-generation models. It mainly requires translating textual content in images and rendering the translated text, rather than demanding deep semantic reasoning across modalities. This means the task only tests surface-level multi-modal alignment, not the complex reasoning ability that unified models should possess .
2. The generation-side evaluation of the IMT task is too limited. It only focuses on whether the translated text is correctly rendered into the image (preserving background and layout) and does not require the model to generate semantically coherent content based on multi-modal understanding. Thus, it fails to test the core generation capability of unified models, which should involve semantic-based creative generation .
3. The task over-reliance on textual translation may bias the evaluation toward models with strong NLP capabilities, while ignoring their visual understanding and generation strengths/weaknesses. For example, a model with poor image context understanding but excellent translation performance could still score well on IMT, misleading the assessment of its overall multi-modal ability .

**Questions:**

Please see weaknesses.

---

> ### Author Response · Authors · 2025-11-21
> **To Reviewer 64m2**
>
> **[Q1] Assessment on Multi-model Synergy**
>
>  We appreciate the reviewer’s thoughtful concern. However, we respectfully argue that the IMT task does require multimodal synergy beyond surface-level alignment. IMT couples semantic understanding with spatial–visual grounding. Unlike conventional OCR+MT pipelines, IMT requires a unified model to jointly 1) recognize text regions in unconstrained layouts, 2) infer their semantic role within the visual context, 3) plan where and how translated content should be inserted, and 4) generate an image consistent with the original scene geometry, typography, shadows, textures, and background integrity. These steps require cross-modal reasoning and structured visual–semantic coordination, not simple text extraction.
>
> **[Q2] Limited Assessment on Generation**
>
>  We agree that IMT is not designed to evaluate creative or open-ended generation. This is a deliberate design decision.
>
> First, IMT is intended as a controlled, ground-truth–supervised multimodal editing task.  Existing multimodal generation benchmarks often rely on subjective human preference ratings or LLM-as-a-judge scoring, which introduces noise and model bias. In contrast, IMTBench provides objectively verifiable ground truth, paired before/after editing images, and decomposable metrics evaluating complementary aspects of multimodal generation. This enables rigorous, reproducible evaluation that creative generation tasks cannot provide. Furthermore, IMT probes visual generation fidelity, not creative capacity. IMT aims to generate translated image with precise placement of translated text, realistic blending with original textures, preservation of geometry, layout, and background, and consistent style and semantics. These are core capabilities of vision–language generation systems, distinct from creative generation but equally fundamental.
>
> **[Q3] Potential over-reliance on textual translation**
>
> This is an important concern, and we agree such bias must be avoided. IMTBench was intentionally designed with three orthogonal metrics to prevent over-reliance on translation quality:
> - $S_{text}$: translation correctness
> - $S_{align}$: visually grounded text placement correctness
> - $S_{vision}$: background/layout fidelity
> These metrics jointly ensure that a model with strong translation ability but weak visual reasoning cannot score well. For example, Models such as Qwen-VL, LLaVA, and GPT-4o-mini exhibit strong NLP performance. However, they fail to accurately render translated text spatially, or distort background textures. This results in large drops in $S_{align}$ and $S_{vision}$, demonstrating that the benchmark is not NLP-dominant.
>
> Therefore, IMTBench evaluates multimodal capability in a balanced manner, avoiding the linguistic bias raised by the reviewer.

---

> > ### Comment · Reviewer_64m2 · 2025-11-28
> >
> > Thanks for the response. I agree that IMT is significantly more challenging than a simple OCR+MT pipeline in terms of visual synthesis. However, my concern regarding the depth of reasoning remains. The "synergy" described in the rebuttal focuses heavily on the consistency of the visual output (e.g., blending text into the scene) rather than the comprehension of the visual narrative. The task principally evaluates the model's ability to act as a high-fidelity editor rather than a reasoner. Consequently, I keep my original rating.

---

### Official Review · Reviewer_aM16 · 2025-10-27

**Soundness:** 2
**Presentation:** 2
**Contribution:** 2
**Rating:** 4
**Confidence:** 3

**Summary:**

The paper introduces Immersive Multimodal Translation (IMT) as a proxy task to evaluate the synergy between image understanding and joint image/text generation. It presents IMTBench (2k items spanning Document/Web/Scene across 9 languages) and IMT-1M for training, and scores systems via three metrics—text quality, edit correctness, and background fidelity—aggregated into a single score. Results show commercial cascaded pipelines lead on text and alignment, while unified models better preserve visual naturalness. Fine-tuning on IMT-1M improves open-source models but they still trail the pipelines.

**Strengths:**

1.Clear task formalization and decomposition; the explicit prohibition on using T_src is consistent with end-to-end evaluation goals.

2.Three non-MLLM metrics with formal definitions; the scoring pipeline is reproducible.

3.Broad scenario and language coverage with transparent dataset statistics and visualizations.

4.Practical fine-tuning setups with hardware disclosure.

**Weaknesses:**

1.Limited methodological novelty. IMT primarily integrates existing TIT/IIT lines and known metrics; no new learning paradigm or loss is introduced.

2.Evaluation fairness. Proprietary systems are evaluated via paired understanding+generation APIs (e.g., GPT-4o + GPT-Image-1), which conflicts with the end-to-end restriction and may bias cross-paradigm comparisons.

3.Metric specification & robustness gaps. COMET version/normalization, OCR engine, and mask-generation details are missing; no human correlation or sensitivity analysis is provided.

4.Scene reference bias. Reference images are created by GPT-Image/SeedEdit, which may anchor S_vision toward those editing styles.

**Questions:**

1.Was train–test de-duplication performed between IMT-1M and IMTBench？

---

> ### Author Response · Authors · 2025-11-21
> **To Reviewer aM16**
>
> **[Q1] Limited methodological novelty**
>
> We thank the reviewer for this observation. IMTBench is intentionally positioned as an evaluation framework, not a training methodology. Its novelty lies in establishing the first unified, scalable, and objectively measurable benchmark for immersive, end-to-end multimodal translation. Specifically, IMTBench provides several contributions that go substantially beyond a simple integration of IIT/TIT tasks.
>
> 1. New Task Formulation: We extend IIT/IIMT into a dual-modality supervised task (IMT), requiring both a translated image and translated text.
>
> 2. Large-Scale, Curated Benchmark IMTBench introduces 9 languages × 3 domains, with high-fidelity ground-truth generation and strict quality control, covering both document-style and complex natural scenes.
>
> 3. Three Complementary Metrics forms the first holistic, interpretable decomposition of multimodal translation performance.
>
> Thus, the contribution of IMTBench lies in evaluation framework construction, task formalization, and paradigm-level analysis, rather than in introducing new network modules or training losses.
>
> **[Q2] Evaluation fairness regarding proprietary systems**
>
> Thank you for pointing out this issue. Proprietary systems currently do not provide a unified end-to-end understanding–generation API, so the only feasible approximation is to pair the understanding and generation models from the same provider (e.g., GPT-4o + GPT-Image-1). This protocol is also used in prior work, such as Holistic Evaluation for Interleaved Text-and-Image Generation [1].
> Following this established practice, we adopt the same pairing strategy solely to simulate the closed-source models’ integrated capability under publicly available APIs. All proprietary systems are evaluated under the same procedure, ensuring fairness across paradigms.
>
> Reference:
> [1] Holistic Evaluation for Interleaved Text-and-Image Generation, https://arxiv.org/pdf/2406.14643.
>
> **[Q3] Missing metric details**
>
> We thank the reviewer for pointing out these omissions, and we will incorporate the missing specifications into the main paper. For COMET configuration, we use COMET-22 (unbabel/wmt22-comet-da) to evaluate. For OCR engine, we use PaddleOCR3.0 with multilingual recognition enabled. For mask generation, we use detection results from PaddleOCR and mask text region. For human correlation and sensitivity analysis, We acknowledge the importance of validating metrics with human judgments. We will further collect additional human preference annotations and compute their correlations with our proposed metrics ($S_{text}$, $S_{align}$, and $S_{vision}$). This expanded human evaluation will be incorporated and continuously improved in our future work.
>
> **[Q4] Scene reference bias**
>
>  This is a valuable concern. We chose GPT-Image-1 and SeedEdit because they represent state-of-the-art high-fidelity editing systems, but we emphasize IMTBench evaluates faithfulness, not stylistic imitation. $S_{vision}$ masks out all text areas. Therefore, evaluation focuses only on background preservation, which is largely style-agnostic. In addition, reference images are used solely as ground truth, not as exemplars to imitate. The model's output is compared to its own matched reference, not to GPT-Image's style.

---

### Official Review · Reviewer_xNUv · 2025-10-31

**Soundness:** 2
**Presentation:** 2
**Contribution:** 2
**Rating:** 2
**Confidence:** 4

**Summary:**

This work proposes In-Image Machine Translation (IIMT) as a way to evaluate unified multimodal models (text and image modalities).
The authors introduce IMTBench, a novel benchmark covering 2,000 samples across 3 splits (scene, web, document) and 9 languages (Arabic, Russian, Chinese, English, Japanese, Italian, French, German, Spanish).
Furthermore, the authors generate 1 million training samples following the same pipeline and show that fine-tuning UniWorld can improve the models’ performance on IMTBench.

**Strengths:**

IIMT is a quite relevant task for the field with multiple widely adopted commercial solutions (such as e.g. Google Translate’s visual translation and equivalents). Yet, only limited image-to-image translation data is available for research.
The introduced IMTBench and IMT-1M training data seem a meaningful contribution to this area of research and improve over existing sets in diversity, scale, and realism.

**Weaknesses:**

* The paper’s opening claims that the IMT task is a more objective evaluation over other interleaved image-text generation evaluations such as InterleavedBench. However, this claim of higher “objectiveness” seems not substantially explored in the paper:
    * First, some relevant prior work such as ISG-Bench [1] or MMIE [2] are not discussed. The claim of the introduction that other benchmarks “can only evaluate performance in a single modality” seems not correct given this prior work (MMIE, ISG-Bench).
    * Secondly, the argument that since those evaluations include a model-as-a-judge would mean that they “cannot be regarded as truly objective metrics” seems to imply that using a model for judging is what makes them not “objective”, yet the proposed IMTBench also uses model based evaluation (COMET, OCR, Mask LPIPS). Moreover, the “Scene” split is generated using GPT-Image and SeedEdit which are then filtered manually for correctness. One may argue that such method will also introduce non-zero bias based on those generation model’s capabilities in the resulting dataset. Such issues are arguably minor but then the claim that the this evaluation will be clearly more objective than judgement with a strong multimodal LLM without further study is perhaps not clearly true.
* In section 4.1, the manuscript mentions that proprietary unified multi-models in the study cannot generate text and images simultaneously but GPT-4o’s native image generation had already been released back in March 2025. Earlier sections also mention the more recent Nano Banana model, which also offers such capabilities. Adding these or other frontier models would make the presented results more relevant.
* In section 5, fine-tuning based on the introduced IMT-1M sample is explored. Section 5.1 discusses training loss curves and attributes the shape and absolute values to differences in the model’s pre-training and architectures. However, it seems that this is based training runs following the models’ original training recipe which are naturally not consistent between these models. I am not sure if such limited study allows a conclusion such as “purely autoregressive architectures are limited by slower convergence, whereas strong diffusion-based pre-training enables faster generalization on the IMT task”. Furthermore, while loss curves are shown for Bagel, Uniworld, and Janus, table 2 only shows performance impact on UniWorld. Why are the results on the other models not reported? The qualitative result in Figure 6 is peculiar since as far as I can assess the French translation, the result after fine-tuning is incorrect and has no meaning in French. This seems to contradict the statement that the model “acquires correct semantic knowledge to produce accurate translations” and may imply that the fine-tuning may have impacted instruction following more than task quality (section 5.2). As such, evidence of the efficacy of fine-tuning on IMT-1M appears limited.
* There is limited study as to where and how models fall short on the tasks introduced in this work.


Minor: the paper has some typos:
* In figure 1, inconsistent spelling: “Single-modal”, “Single Modal”
* In section 1: “Nano Banana” is written as “Banana-nano”
* In section 4.1: “Findings” (upper case F when it should be lower case)


Overall, I feel this work’s main contributions are a novel dataset for the somewhat established IIMT task, which additionally includes a text ground truth. The proposed evaluation protocol seems defined based on a desire to reduce the reliance on model-as-a-judge evaluations. While the advantage of the proposed protocol is not directly studied, the protocol itself appears reasonable.
The writing appears to somewhat over-emphasis the novelty beyond these (already meaningful) contributions.


[1] Chen, Dongping, et al. "Interleaved scene graphs for interleaved text-and-image generation assessment." arXiv preprint arXiv:2411.17188 (2024).
[2] Xia, Peng, et al. "Mmie: Massive multimodal interleaved comprehension benchmark for large vision-language models." arXiv preprint arXiv:2410.10139 (2024).
[3] Gao, Yu, et al. "Seedream 3.0 technical report." arXiv preprint arXiv:2504.11346 (2025).

**Questions:**

* In section 2.2, the authors describe the task they call “IIT”. However, in the cited literature, this task seems to be called “TATI” (Qian et al., 2024) and “IIMT” (Lan et al, 2024, Tian et al., 2025a, Tian et al., 2025b). Is there a reason for the difference in naming?
* The introduced “IMT” task seems a combination of IIMT with outputting the translated text (as text) as described in 3.1. However, this is not clearly marked as the definition of “IMT”. Is my understanding correct? Can this be clarified in the text? (Additionally, in section 2.2 IIT is described as “directly replace the source language text within the image with the corresponding target language text, without generating textual output as an intermediate result”; Is this strictly true? While IIT, at least IIMT, may not test any intermediate text being generated I am not sure if generating such text would strictly speaking change the task? This may even happen transparently with e.g. prompt rewriting/expansion in some systems.)
* What OCR engine is used for S_align? Can this be documented in the paper to ensure reproducibility?
* For S_vision, can you elaborate how the binary mask works, exactly? The annotations shown in Figure 12 do not seem to show mask annotations? Also, wouldn’t this mean the method would not reward preserving text style / font? Has an unmasked version of the metric been considered?
* From [3], Seedream 3.0 supports only Chinese and English, yet it seems it was tested against IMTBench which covers 9 languages. Would it make sense to report performance on officially supported languages separately, perhaps in the appendix? Are the trends the same as when averaged across all locales? While Figure 3 provides some radar plots in this direction, it is my understanding that this still represents 9-1 and 1-9 averages, which do not allow this analysis?
* Nit / suggestion: In Figure 4, the target ground truth images are not provided, adding them may make the figure a bit more comprehensive?

---

> ### Author Response · Authors · 2025-11-21
> **To Reviewer xNUv**
>
> **[Q1] Naming inconsistency**
> We thank the reviewer for highlighting this terminological inconsistency.  Indeed, recent literature uses multiple names for similar tasks:
> - TATI (Translate AnyText in Image) – Qian et al., 2024
> - IIMT (In-Image Machine Translation) – Lan et al., 2024; Tian et al., 2025a,b
> - IIT (In-Image Translation) – terminology we adopted in Section 2.2
>
> Our use of “IIT” reflects a generalized umbrella term describing the family of tasks whose goal is to directly replace source-language text inside images with target-language text. We intentionally adopted this simpler, modality-agnostic name to unify the terminology, as the community has not yet converged on a single standard. Nevertheless, we agree that aligning our terminology with the cited works would improve clarity. In the revision, we will explicitly state that IIT corresponds to the TATI/IIMT tasks as named in prior literature, and briefly explain the rationale for using a generalized name.
>
> **[Q2]  Does IIT strictly forbid intermediate text generation?**
>
> We also agree with the reviewer, while IIT/IIMT tasks do not evaluate textual intermediate outputs, they do not strictly prohibit models from internally generating text. For example, Translatotron-V uses OCR loss and translation loss to optimize IIT model, which proves it can output intermediate  OCR and translated result. However, during the evaluation stage on IIT, it only requires model output is an edited image.  We will refine the wording in Section 2.2 to avoid implying a strict prohibition and instead emphasize that IIT does not expose textual output to the evaluation protocol, which is the essential distinction.
>
> **[Q3]  OCR engine**
>
>  We indeed use PaddleOCR3 as the OCR engine for $S_{align}$. We will add this information in the next version.
>
> **[Q4]  The binary mask used in $S_{vision}$**
>
> For the binary mask used in $S_{vision}$, we use detection results from PaddleOCR and mask text region. he binary mask marks background pixels as 1 and text/edited pixels as 0. $S_{vision}$ evaluates only background preservation by ignoring intended textual modifications. We will add this clarification explicitly in the paper.
>
> In addition, $S_{vision}$ intentionally ignores font or glyph preservation. The rationale is:
>
> - Style/glyph quality is already measured indirectly by Salign, since OCR errors strongly correlate with distorted or unnatural text appearance.
> - Including text regions in LPIPS would undesirably penalize correct translation or legitimate visual differences between scripts (e.g., Translate Chinese texts into Arabic).
>
> We experimented with unmasked LPIPS early in development. However, it proved unsuitable. Because it strongly penalized correct text edits, and favored pipelines that simply “erase” text or hallucinate blurry textures, which contradicts the evaluation goals. Thus, masked LPIPS is necessary to isolate background fidelity, a core aspect of immersive multimodal generation.
>
> **[Q5] Seedream 3.0 language support**
>
> This is an excellent point. Seedream 3.0 officially supports only Chinese and English, but we evaluate it on all 9 IMTBench languages for completeness. Because our benchmark compares architectural paradigms (unified models vs. cascaded pipelines) and the lack of multilingual support is itself an important limitation for unified systems.
>
> **[Q6] Suggestion for Figure 4**
>
> We thank for this useful suggestion. We will revise Figure 4 to include the reference translated images for better completeness and readability.

---

### Official Review · Reviewer_avao · 2025-11-03

**Soundness:** 2
**Presentation:** 3
**Contribution:** 2
**Rating:** 2
**Confidence:** 3

**Summary:**

The paper identifies a key challenge in evaluating unified multimodal models: existing benchmarks either assess generation and understanding tasks in isolation or rely on subjective, potentially biased scoring from other large models (e.g., GPT-4). To address this, the authors propose a new proxy task called Immersive Multimodal Translation (IMT). The task requires a model to take an image containing text, translate that text into a target language, and render the translated text back into the image while preserving the background.
To support this task, they introduce IMTBench, a new benchmark with 2,000 samples across documents, webpages, and scenes, spanning nine languages. They also propose a three-part evaluation framework based on objective, established metrics: (1) translation quality using COMET (Stext), (2) text rendering fidelity using OCR accuracy (Salign), and (3) background preservation using masked LPIPS (Svision). Their experiments show that specialized commercial pipelines significantly outperform current open-source and proprietary unified models on this task, suggesting a large gap for future work.

**Strengths:**

1. Well-Motivated Problem: The paper correctly identifies a significant issue in the field: the over-reliance on subjective, expensive, or biased MLLM-based scorers for evaluating generative multimodal models. The goal of creating an objective, automated evaluation is highly laudable.

2. Significant Data Collection Effort: The creation of IMTBench and the associated 1M-sample training set is a substantial engineering effort. Curating a multilingual, multi-domain dataset for this task is non-trivial and provides a new resource for the community.

**Weaknesses:**

1. The core of the proposed evaluation is insufficient. The $S_{align}$ score (OCR accuracy) only measures if the translated text is legible, not if it is rendered believably. Key visual aspects like font style, color, lighting, perspective, and blending—all crucial for an "immersive" result—are completely ignored. This is a major gap that undermines the claim of a comprehensive evaluation framework.

2. The central result from the experiments is that specialized commercial APIs (which are likely multi-stage pipelines of OCR, machine translation, and inpainting models) outperform end-to-end unified models. This is largely expected. Specialized tools are almost always better than general-purpose ones on their specific task. This finding does little to advance our fundamental understanding of unified model architectures.

3. A large portion of the benchmark (Document, Web) and even the ground truth for the "real-world" Scene images are generated synthetically. This limits the benchmark's ability to evaluate performance on the vast and unpredictable diversity of real-world typography and image conditions.

4. Could you elaborate on the novelty of the IMT task compared to the existing body of work on scene text translation and editing? What makes IMT a better "proxy task" for general multimodal intelligence than these related problems?

**Questions:**

While the paper is well-written and addresses an important problem, its core contributions are minor. The proposed evaluation protocol is insufficient for the very task it defines, as it might ignore the visual quality of the text generation. Please find the questions in the weakness part above, and hopefully the authors could address them.

---

> ### Author Response · Authors · 2025-11-21
> **To reviewer avao**
>
> **[Q1] About evaluation on IMT**
>
> We respectfully believe that the IMT task captures essential aspects of cross-modal synergy rather than surface-level alignment.
>
> First, IMT requires tightly coupled bidirectional reasoning between understanding and generation. Unlike traditional OCR+translation+editing pipelines, unified models must interpret layout, typography, spatial structure, and scene semantics while generating a visually coherent image that embeds the translated content. This joint preservation of linguistic meaning and visual context demands deeper multimodal reasoning than text-only translation.
>
> Second, IMT evaluates context-aware transformation rather than isolated text substitution. Many cases involve semantic dependencies between text and surrounding visual elements (icons, UI components, scene objects). Maintaining semantic coherence with the background cannot be achieved without modeling these higher-level visual–linguistic relationships.
>
> Finally, we view IMT as an early step toward holistic visual text generation. Existing text-editing benchmarks do not assess visual properties such as font style, color, lighting, perspective, and blending. We believe more comprehensive evaluation of these aspects will further enrich IMT and appreciate the reviewer’s feedback for pointing us in this direction.
>
> **[Q2] Comparison with commercial API**
>
> We agree that commercial multi-stage pipelines outperform unified models, which is expected. Our contribution, however, is to provide the first systematic characterization of why unified end-to-end models lag behind and where their fundamental weaknesses lie.
>
> Existing benchmarks do not jointly evaluate linguistic fidelity, visual coherence, and cross-modal consistency. IMT reveals consistent failure pattern, including semantic drift, background–text misalignment, and font rendering issues, highlighting structural limitations in current unified architectures.
>
> Unified LVLMs do not explicitly perform OCR, translation, and editing; they must implicitly integrate these stages. The performance gap clarifies which components (vision encoding, text replacement, visual blending, layout preservation) currently bottleneck unified solutions and require architectural innovation.
>
> Our benchmark tracks progress as unified models evolve. Commercial pipelines serve as upper bounds, not competitors. The scientific value lies in quantifying the gap and identifying the multimodal reasoning capabilities that unified architectures must strengthen.
>
> Thus, our work contributes to the fundamental understanding of unified multimodal models by making their limitations measurable, interpretable, and comparable, which we believe is crucial for advancing this research direction.
>
> **[Q3] Generating images of IMTBench Synthetically**
>
> We thank the reviewer for the comment. While parts of IMTBench are synthetic, this design is both necessary and faithful to real-world IMT scenarios. For the Web category is rendered from real HTML pages, producing faithful reproductions of actual webpages that naturally reflect real-world layout and typography. For the Document category uses the SynthDoG engine with natural backgrounds, perspective distortion, blur, and noise, closely matching real document photography. Such rendering is widely used in prior works (e.g., Qwen-Image) and captures realistic variability in document translation tasks. For Scene images, collecting large-scale paired IMT data is practically infeasible due to the need for scene-level text detection, erasure, human translation, and realistic inpainting. Closed-source commercial models currently offer the only scalable way to generate paired data while preserving geometric, lighting, and material consistency. These synthetic but physically grounded transformations effectively simulate real open-world IMT conditions.
>
> **[Q4]  Novelty of IMT task**
>
> IMT differs substantially from existing scene text translation and editing tasks and serves as a stronger proxy for general multimodal intelligence.
>
> First, IMT unifies visual language alignment, semantic translation, and visual reintegration into a single end-to-end objective. Prior tasks treat these stages separately. IMT requires joint reasoning over linguistic content, layout, typography, and background semantics, demanding more integrated multimodal capability.
>
> Second, IMT is context-dependent rather than patch-local. Existing editing tasks operate on small regions and assume local independence. IMT instead requires preserving global scene coherence—semantic relations between text and objects, functional consistency in webpages, and stylistic harmony in documents—linking high-level perception with generation.
>
> Third, IMT evaluates capabilities largely missing from current benchmarks, such as cross-modal consistency, layout preservation, long-text fidelity, and visual blending quality. These are exactly the abilities emerging in modern LVLMs but remain under-assessed elsewhere.

---

### Meta-Review · Area_Chair_QRoD · 2026-01-05

**Summary:**

Summary of the reviewer's concerns:
1. The evaluation metrics are too narrows (mentioned by 64m2, avao) The paper leverages automatic evaluations to eval whether the translated text is legible, while this task ignores evaluating models with multimodal reasoning capability. This evaluation coverage is not wide enough.
2. The images in the benchmark are generated synthetically, which might not cover the full real world scenarios. Also the images are generated by using the GPT-Image / SeedEdit, which might bring the bias to the evaluation (mentioned by 64m2, avao, aM16)
3. The motivation of this paper is proposing a dataset that reduces the model-as-a-judge bias, while the proposed metrics are still model-based metrics. Those two seems violate each other (xNUv, aM16)
4. Reviewer avao mentioned that the the benchmark and the analysis done by this paper didn't reveal the direction for the future improvements.

**Reviewer Concerns:**

1. For the concern 1, the rebuttal claimed that 'the IMT task does require multimodal synergy beyond surface-level alignment. IMT couples semantic understanding with spatial–visual grounding. Unlike conventional OCR+MT pipelines, IMT requires a unified model to jointly 1) recognize text regions in unconstrained layouts, 2) infer their semantic role within the visual context, 3) plan where and how translated content should be inserted, and 4) generate an image consistent with the original scene geometry, typography, shadows, textures, and background integrity. These steps require cross-modal reasoning and structured visual–semantic coordination, not simple text extraction.' However the metrics proposed by the paper only captured translation quality,  text fidelity and background preservation. They didn't evaluate the capabilities mentioned in the rebuttal (from step 1 to step 4)
2. For the concern 2, the rebuttal claimed that the synthetic but physically grounded transformations effectively simulate real open-world IMT conditions and also mentioned several prior works for using the same / similar approach for generating the synthetic images. However, this rebuttal only address the approach for generating the images is reasonable, but didn't address the concern of the coverage.
3. For the concern 3, the rebuttal didn't touch that concern.
4. For the concern 4, the rebuttal claimed that this paper contributes on the fundamental understanding of unified multimodal models, however, without explicitly reasoning chain to support this.

**Reviewer Scores:**

I've read the full rebuttal.
1. I think the reviewer avao's main concern 'The proposed evaluation protocol is insufficient for the very task it defines, as it might ignore the visual quality of the text generation' is not fully addressed.
2. Reviewer xNUv's concern 'The writing appears to somewhat over-emphasis the novelty beyond these (already meaningful) contributions.' (also the weakness 1) is not fully addressed.
3. Reviewer aM16's concern should be answered by the rebuttal except the one mentioned in the reviewer concerns's part. Likely the reviewer might improve the rating.
4. Reviewer 64m2 stated that the rating is maintained.

Given the above summarization, I don't think the reviewer scores would be substantially improved after the rebuttal.

---

### Decision · Program_Chairs · 2026-01-26

Reject